# Stealth Coating of Nanoparticles in Drug-Delivery Systems

**DOI:** 10.3390/nano10040787

**Published:** 2020-04-20

**Authors:** See Yee Fam, Chin Fei Chee, Chean Yeah Yong, Kok Lian Ho, Abdul Razak Mariatulqabtiah, Wen Siang Tan

**Affiliations:** 1Department of Microbiology, Faculty of Biotechnology and Biomolecular Sciences, Universiti Putra Malaysia, Serdang 43400, Malaysia; cyee531@hotmail.com (S.Y.F.); yongcheanyeah@hotmail.com (C.Y.Y.); 2Nanotechnology and Catalysis Research Centre, University of Malaya, Kuala Lumpur 50603, Malaysia; cheechinfei@um.edu.my; 3Department of Pathology, Faculty of Medicine and Health Sciences, Universiti Putra Malaysia, Serdang 43400, Malaysia; klho@upm.edu.my; 4Department of Cell and Molecular Biology, Faculty of Biotechnology and Biomolecular Sciences, Universiti Putra Malaysia, Serdang 43400, Malaysia; mariatulqabtiah@upm.edu.my; 5Laboratory of Vaccines and Immunotherapeutics, Institute of Bioscience, Universiti Putra Malaysia, Serdang 43400, Malaysia

**Keywords:** nanoparticles, polymer, stealth, drug delivery, opsonization, phagocytosis

## Abstract

Nanoparticles (NPs) have emerged as a powerful drug-delivery tool for cancer therapies to enhance the specificity of drug actions, while reducing the systemic side effects. Nonetheless, NPs interact massively with the surrounding physiological environments including plasma proteins upon administration into the bloodstream. Consequently, they are rapidly cleared from the blood circulation by the mononuclear phagocyte system (MPS) or complement system, resulting in a premature elimination that will cause the drug release at off-target sites. By grafting a stealth coating layer onto the surface of NPs, the blood circulation half-life of nanomaterials can be improved by escaping the recognition and clearance of the immune system. This review focuses on the basic concept underlying the stealth behavior of NPs by polymer coating, whereby the fundamental surface coating characteristics such as molecular weight, surface chain density as well as conformations of polymer chains are of utmost importance for efficient protection of NPs. In addition, the most commonly used stealth polymers such as poly(ethylene glycol) (PEG), poly(2-oxazoline) (POx), and poly(zwitterions) in developing long-circulating NPs for drug delivery are also thoroughly discussed. The biomimetic strategies, including the cell-membrane camouflaging technique and CD47 functionalization for the development of stealth nano-delivery systems, are highlighted in this review as well.

## 1. Introduction

Over the past few decades, nanotechnology has contributed tremendously to the advance and development of nanoscale materials and nanoparticles (NPs) for various biomedical applications. NPs have received considerable interest in the delivery of therapeutic agents owing to their unique features of large surface-to-volume ratio with a greater capacity for drug loading and high functionalization possibilities [1]. Undoubtedly, the use of therapeutic drugs in clinical applications is often hindered by their intrinsic drawbacks including poor solubility as well as adverse pharmacokinetics and biodistribution [2]. Yet, NPs that serve as drug reservoirs can potentially improve the drug solubility and prolong the blood circulation half-life, releasing the drug in a controlled and sustained manner to minimize the systemic side effects and further improve the pharmacokinetics [3]. The encapsulation and attachment of therapeutic cargos such as chemotherapeutic drugs, peptides, nucleic acids, and ligands using NPs can enhance the efficacy of targeted delivery to specific tumor cells with fewer adverse effects [4]. Several classes of nanomaterial including synthetic materials such as liposomes [5], micelles [6], dendrimers [7], hydrogels [8], and natural biomaterials such as virus-like nanoparticles [9], are widely employed as smart drug-delivery systems for cancer therapies.

Regardless of the therapeutic purpose of NPs, a prolonged circulation is a requisite for effective drug delivery and therapeutic efficacy. Despite the remarkable features of NPs, their applications as nanocarriers are often restricted by the potential antigenicity and immunogenicity that lead to the rapid clearance of NPs due to their interactions with the mononuclear phagocyte system (MPS) or complement system [10]. This resulting premature NP elimination from blood circulation will cause the therapeutic agent to be released at off-target sites, compromising drug-delivery efficacy to tumor cells. Upon the administration into the bloodstream, unprotected NPs tend to be recognized by opsonins such as the immunoglobulins, complement proteins, or receptors present on the surface of macrophage plasma membrane [11]. Consequently, phagocytic cells mark the opsonized NPs for uptake and elimination before reaching the tumor cells for drug delivery. Therefore, it is of utmost importance to obtain an in-depth understanding of the interactions between NPs and their complex biological environments that affect their recognition and elimination by the immune system.

For targeted drug delivery, the blood circulation time of NPs is a critical factor that determines the NPs’ accumulation at the target site for an effective drug release [12]. Hence, by applying a camouflaging technique on NPs, the blood circulation half-life of the nanomaterials can be improved by escaping the recognition and clearance by the MPS and/or complement system [13,14,15]. This can be achieved by grafting a stealth-coating layer onto the surfaces of NPs, restricting the interactions between NPs and opsonin proteins that mediate the phagocytic clearance [16,17,18]. This stealth delivery system commonly recruits polymers such as the gold standard poly(ethylene glycol) (PEG) to impart MPS-avoidance characteristics to NPs [19]. In this review, we provide an overview of the state-of-the-art in the development of several polymers and biomimetic materials for stealth coating of NPs in drug delivery systems. Moreover, the fundamental polymer coating properties influencing the stealthiness of NPs, which include their molecular weight, density, and surface conformation, will also be discussed thoroughly.

## 2. Opsonization and Phagocytosis

The controlled delivery of therapeutic agents to tumor cells using nanocarriers is essential for efficient drug delivery to reduce the systemic toxicity and improve the pharmacological profiles. In order for NPs to remain in the bloodstream until they reach the therapeutic site, they must be capable of evading phagocytic capture by monocytes and macrophages of the MPS, previously known as the reticuloendothelial system (RES) [20]. Thus, the opsonization or removal of nanocarriers from the bloodstream by the MPS is a major challenge to the realization of effective drug delivery. Most of the unprotected NPs, regardless of their materials of composition, can be removed from the blood circulation by the MPS within seconds to minutes upon intravenous administration [13]. In fact, the macrophages are not able to capture the NPs themselves, but they can recognize the opsonin proteins bound to the surface of NPs [21]. Opsonins are any blood-circulating components including immunoglobulins, blood clotting factors, and complement proteins [12,21]. They play a crucial role in phagocytic recognition due to the charge repulsion resulted from the negative charges of both bacteria/viruses and phagocytes [22].

Circulating proteins rapidly adsorb to the surface of NPs, forming a protein corona for various biological events including particle opsonization, phagocytic clearance, formation of immune complexes, and activation or suppression of immune system [23,24]. Opsonization is the process of opsonin adsorption onto the surface of NPs following recognition, to render it viable for phagocytic cells. Subsequent phagocytosis will initiate engulfment, destruction, and eventually removal of the NPs from the bloodstream. In other words, the phagocytic clearance of NPs is mediated by opsonization. However, the mechanism of opsonization is not fully elucidated due to complicated activation pathways, but the crucial components involved are rather well-known. Immunoglobulins and complement proteins such as C3, C4, and C5 are the most common opsonins responsible for the rapid clearance of nanoparticulate drug carriers [21]. Nonetheless, the interactions between opsonins and NPs depend on the size and surface characteristics of the NPs [25,26,27]. It is thought that the opsonins come into contact with NPs by random Brownian motion. Upon interaction with NPs, other attractive forces including van der Waals, electrostatics, ionic, hydrophobic, hydrophilic, and/or hydrogen bonding are also involved in opsonins binding to the surfaces of NPs [28,29].

The subsequent step following opsonization is the phagocytic clearance, in which the phagocytes are attached to the NPs via surface-bound opsonin proteins. The opsonins typically undergo a conformational change that can be captured by the recruited phagocytes via their specialized receptors, thereby alerting the phagocytes to the presence of foreign particles [30]. When opsonins encounter NPs with strong hydrophobicity, phagocytosis occurs due to the non-specific phagocytes binding to the surface-bound serum proteins. However, the most significant phagocyte attachment method following opsonization is the complement activation, a key component of innate immunity that can be activated via three main pathways: classical, alternative, and lectin pathways. The complement system comprises about 30 soluble and membrane-bound proteins, either possessing enzymatic and binding functions or presenting as receptors on the surface of immune cells [31].

A schematic overview of three activation pathways of the complement system is outlined in Figure 1. Regardless of the activation pathway, the enzymatic cascade generates C3 convertase, which cleaves the most abundant C3 protein, into C3b and C3a. As the key effector molecule of the complement system, C3b and its inactive fragment iC3b bind covalently to the cell membrane and opsonize the antigen for eventual phagocytosis. Subsequently, C3b binds to C3 convertase to form C5 convertase that generates C5b and C5a. The assembly of the terminal complement proteins (C5b, C6, C7, C8, C9) contributes to the formation of the membrane attack complex (MAC) that disrupts the cellular lipid bilayer of the extracellular materials including bacteria, viruses, and nanocarriers. Meanwhile, C3a and C5a act as potent inflammatory mediators [11]. The detailed mechanisms of the complement system are not within the scope of this paper, yet there are several excellent literature reviews focusing on this subject [21,32,33,34].

Following phagocyte attachment, the final step is the ingestion of extracellular materials through endocytosis whereby the enzymes and oxidative-reactive chemical factors secreted by phagocytes will break down the phagocytosed materials [35]. Depending on the molecular weight and relative size, most of the non-biodegradable NPs will either be removed via renal filtration or sequestered in the MPS organs if their molecular weight is beyond the renal threshold. The initial opsonization is a crucial step for subsequent phagocytic recognition and clearance from the blood circulation. While there are no definite rules and approaches to inhibit completely opsonization, tremendous research over the last few decades has found that grafting a stealth polymer coating on NPs is effective in suppressing the interaction between opsonins and NPs, leading to prolonged blood circulation half-life and increased efficacy of drug deliveries [22].

## 3. Polymers for Stealth Functionalization

As compared to hydrophilic and neutrally charged particles, hydrophobic and charged particles tend to be opsonized more easily in the bloodstream [36,37,38,39]. Therefore, it is recognized that long-circulating NPs can be achieved by grafting a hydrophilic polymer that prevents the opsonization for stealth-delivery systems. Consequently, prolongation of NPs in the blood circulation can be improved from a few minutes to several hours, thereby increasing drug-delivery efficacy to tumor sites [12,40]. Amongst the numerous stealth polymers, the gold standard poly(ethylene glycol) (PEG) has received widespread interest as the polymer of choice to develop stealth nanocarriers for drug delivery over the past decades. This is attributed to their flexible and hydrophilic nature that sterically impedes NPs from interacting with the surrounding opsonin proteins, thereby resulting in increased blood circulation half-life [30]. In short, the polymers that endow nanocarriers with stealth behaviors typically exhibit a few common features, namely high hydrophilicity and high flexibility [22]. In addition to PEG, other potential alternative polymers such as poly(2-oxazoline) (POx) and poly(zwitterions) that confer stealth properties to NPs [41], are also described in the subsequent sections. The chemical structures of the stealth polymers are presented in Figure 2.

### 3.1. Poly(Thylene Glycol) (PEG)

Undoubtedly, PEG has received considerable interest for the development of stealth coating on the surface of NPs particularly in drug-delivery system owing to their high hydrophilicity properties with low toxicity in vivo. The term PEGylation refers to the grafting of PEG chains on the particle surface [42]. Apart from covalent conjugation through surface functional groups, PEG chains may be covalently attached to the surface of NPs using PEG derivatives such as poly(alkylcyanoacrylates), poly(lactic acid) (PLA), and poly(lactic acid-*co*-glycolic acid) (PLGA) [43,44]. The surface features of hydrophilic grafting, particularly PEG, have been described by Owens and Peppas [30]. The hydrophilic and flexible nature of PEG chains bestow a hydrated cloud as well as an extended conformation in solution. When the opsonins encounter the PEGylated NPs, the surface-bound opsonins begin to compress the extended PEG chains, shifting to a higher energy conformation. The opposing repulsive force resulting from this change in conformation can completely suppress the attractive force between opsonin proteins and the surface of NPs [30]. This protein repulsion resulting from PEG was also visualized by Peracchia et al. [45] using freeze-fracture transmission electron microscopy (TEM).

The most widely accepted theory on the stealth properties of PEG is based on the interactions between opsonins and the PEGylated surface. This theory supports the hypothesis that the PEG chain coating exhibits an extended and flexible surface barrier layers that can suppress the adhesion of opsonin proteins by allowing them to be “invisible” for phagocytic capture [46]. Yet, in order to achieve an effective blocking of opsonins to nanocarriers, it is a necessity to exceed a minimum layer thickness of polymer surface coating that correlates to several critical factors such as the molecular weight, surface chain density as well as polymer conformation. These contributing factors of stealth-coating effectiveness will be discussed in the subsequent sections of this review.

Ever since the introduction of Doxil^®^ in 1995, the first Food and Drug Administration (FDA)-approved liposomal cancer therapeutic that encapsulated doxorubicin with PEG, numerous PEGylated products have emerged in the market for various biomedical applications [47,48]. In terms of biomedical aspects, PEGylation as a well-studied surface modification is known to be biocompatible and non-immunogenic while improving blood circulation and reducing NPs uptake by MPS cells [49]. In a study by Lipka et al. [50], it was observed that an enormous amount (more than 95%) of gold nanoparticles modified with 10 kDa PEG (Au-PEG10k) were circulating in the blood one hour after intravenous injection. By contrast, the prolonged circulation time was not observed for Au NPs modified with 750 Da PEG (Au-PEG750). Moreover, the biodistribution study of NPs’ uptake by the RES also demonstrated that approximately 90% of the uncoated Au NPs and Au-PEG750 were captured in the liver and spleen after one hour, while only less than 5% and 54% of Au-PEG10k was detected after 1-h and 24-h intravenous injections, respectively.

Considering the combined merits of NPs and liposomes for improved encapsulation efficiency and sustained drug release, multilayer PLGA-lecithin-PEG-biotin encapsulating doxorubicin nanoparticles (DOX-PLPB-NPs) were designed by Dai et al. [51] for tumor-targeting cancer treatment. The cellular uptake study demonstrated that DOX-PLPB-NPs exhibited a higher doxorubicin accumulation in the tumor cells with stronger in vivo antitumor ability. Instead of using PLGA copolymer, a recent comprehensive in vivo biodistribution study by Shalgunov et al. [52] who developed a drug delivery vehicle based on PLA-PEG NPs encapsulating a model active pharmacological ingredient (API), vincristine. An apparent correlation between the rate of API release and its pharmacokinetics in plasma was shown: controlled changes in the release rates of the encapsulated API from NPs led to gradual alterations of its circulation half-life. Additionally, the authors highlighted that the systemic toxicity resulted from API is associated with the biodistribution of both API and NPs, where the administered doses of API endow an effect for therapeutic purpose.

To date, PEGylation remains a benchmark for the development of stealth nanocarriers in drug-delivery systems. Despite their stealth barrier that reduces protein adsorption, studies have reported that some PEGylated products can generate anti-PEG antibodies [53,54], which contradict the claim that PEG is non-immunogenic [49]. In addition, an unexpected immunogenic response commonly referred to as the “accelerated blood clearance” (ABC) phenomenon is correlated with the rapid clearance of multiple administrations of the PEGylated NPs [53,54,55,56]. Interestingly, the innate existence of anti-PEG antibodies in 20–25% of normal individuals who have never been treated with PEGylated therapeutics has also been reported [57,58,59]. Therefore, these critical shortcomings of PEG have drawn widespread attention and fueled the search for potential alternative stealth coatings for efficient targeted drug-delivery systems to improve cancer therapeutic efficacy.

### 3.2. Poly(2-Oxazoline) (POx)

Poly(2-oxazoline) (POx) is a hydrophilic polymer that has been known for half a century [60]. It is highly tunable through living cationic ring-opening polymerization (CROP) synthesis of the corresponding 2-oxazoline monomers, in which this technique provides an excellent control with respect to POx chain length, molar mass distribution, as well as the chain-end functionality [61]. In recent years, the increasing concern regarding the use of PEG and its potential immunogenicity has led to a renewed interest towards POx as a promising alternative polymer [62,63]. POx is a non-toxic polymer with similar stealth properties as PEG, and it offers numerous advantages such as excellent biocompatibility, thermo-responsiveness, and high stability with a range of end-group and side-chain functionalization [64,65]. Even though the applications of POx in biomedical field arose only recently, two POx polymers, namely poly(2-methyl-2-oxazoline) (PMeOx) and poly(2-ethyl-2-oxazoline) (PEtOx), are widely studied for drug carriers in polymeric therapeutics [65].

POxylation, the coating of POx chains on the surface of NPs, can improve the circulation time of the particles in the bloodstream due to their hydrophilic nature as seen in PEGylation. Chapman et al. [66] reported that the stealth effect of POx is attributed to the absence of hydrogen bond donors in the polymer, which renders them to be highly biocompatible while exhibiting non-fouling behaviors. In addition, in vivo studies also revealed that POx is non-immunogenic even after repeated intravenous and subcutaneous administrations, making POx a potential alternative polymer for stealth drug delivery [67,68]. Besides NPs’ conjugation, POx can also be conjugated with peptides, proteins, and drugs for polymer therapeutics [69,70,71]. This is attributed to its high functionalization possibilities where both the alpha and omega termini of POx can be functionalized by a selection of initiators and terminating agents using the CROP synthesis technique [61].

A major breakthrough for POx in biomedical applications was certainly the first-in-human study of POx-rotigotine conjugate initiated in 2015 [67]. The study is ongoing but the preliminary results are rather promising for the treatment of Parkinson’s disease [67]. Another interesting study by Tong et al. [72] who developed an advance nanovehicle by conjugating superoxide dismutase 1 (SOD1) to POx block copolymers, for delivery across the blood–brain barrier and in neurons. The resulting SOD1-POx conjugates significantly enhanced the neuronal uptake and delivery in mice with negligible cytotoxicity, and augmented plasma circulation half-life as compared to the native and PEGylated SOD1. In addition, these conjugates could cross the intact blood–brain barrier and remained in the brain capillary and parenchyma, but neither native nor PEGylated SOD1 reached the brain. Therefore, these promising results of SOD1-POx conjugates indicate the therapeutic potential of POxylated SOD1 in the treatments of superoxide-related brain diseases.

Recently, Bludau et al. [73] investigated the in vivo circulation of the plant viral nanoparticle, tobacco mosaic virus (TMV) conjugated with PMeOx. It was demonstrated that the POxylated TMV NPs were better shielded from antibody recognition as compared to PEGylated TMV NPs due to a higher density of polymer coating, as assessed by enzyme-linked immunosorbent assay (ELISA). Additionally, the POxylated NPs also exhibited a reduced macrophage uptake compared to the PEGylated NPs. Similarly, an antigenicity study by Fam et al. [74] showed that the covalent conjugation of amine-functionalized PEtOx to hepatitis B core antigen virus-like nanoparticles (HBcAg VLNPs) significantly reduced the antigenicity of HBcAg VLNPs as compared with the unshielded VLNPs. Nonetheless, the ELISA analysis revealed that both POxylated and PEGylated HBcAg VLNPs showed a comparable shielding effectiveness against antibody recognition. This could be attributed to other crucial contributing factors such as polymer conformations on the surface of NPs and the polymer’s chain length that results in a difference of molar mass distribution.

Despite the lack of FDA approval that hinders future research and developments, POx remains as a promising polymer for stealth drug delivery due to its high synthetic versatility and superior properties. Yet, in order to evaluate the potential of POx as an alternative to PEG, further studies are required to ascertain the detailed immune interactions and biodistribution of POx in the bloodstream.

### 3.3. Poly(Zwitterions)

Poly(zwitterions) such as poly(sulfobetaine) (PSB) and poly(carboxybetaine) (PCB) are polyelectrolytes consisting of zwitterionic moieties as monomers, in which the zwitterions contain both positively and negatively charged groups, but with an overall neutral charge [75]. In addition to internally balanced surface charge, these zwitterionic materials possess a superior ionic solvation with water molecules, creating highly hydrophilic surfaces to reduce protein adsorption [76]. Similar to other hydrophilic stealth polymers, zwitterionic polymers can escape the immune surveillance, and also increase the blood circulation half-life [77]. A comparison between PEG and zwitterionic PCB as stealth coatings of NPs for drug delivery has been reviewed by Cao and Jiang [78]. Despite the numerous benefits, it is thought that the amphiphilic nature of PEG will affect the stability of the PEGylated NPs. Zwitterionic materials, in contrast, resolve PEG limitations owing to their super-hydrophilic nature. Based on this evidence, numerous zwitterionic polymers have been widely synthesized as stealth candidates for drug delivery system [79,80].

Liu et al. [81] demonstrated that gold nanoparticles (Au NPs) modified by mixed charged zwitterionic self-assembled monolayers exhibited an excellent plasma protein resistance. In addition, these zwitterionic Au NPs showed a decreased uptake by phagocytic macrophages as compared to both positively and negatively charged Au NPs. In a study by Yuan et al. [82], a charge switchable nanoparticle was developed based on a zwitterionic polymer by incorporating a tumor extracellular acidity sensitive group as the anionic part of the polymer. The resulting zwitterionic polymer-based NPs were demonstrated to reduce the non-specific protein adsorption, and prolong the blood circulation time at the physiological condition. Upon their accumulation in the acidic tumor tissues by the enhanced permeability and retention effect, the NPs were activated to be positively charged by eliminating the anionic part and became recognizable to tumor cells, resulting in an enhanced cellular uptake in vivo, and the inhibition of tumor growth. These zwitterionic polymer-based NPs with tunable surface charge properties have a great potential to achieve cancer therapeutic efficacy.

Zwitterions have become promising candidates in the field of drug delivery due to numerous advantages such as excellent hydration, non-fouling properties, and extended blood circulation time in vivo. However, the research of zwitterionic polymers is hampered as they are insoluble in most of the organic solvents due to their super-hydrophilic nature. Although appreciable research has been made, zwitterionic materials are still in their infancy, particularly at the stage of proof of concept.

## 4. Biologically Inspired Stealth Strategies

Surface engineering is now a pivotal approach to improve the half-life of nanocarriers for drug delivery. Apart from surface modifications through hydrophilic stealth polymer coatings, alternative biologically inspired materials such as cell-membrane camouflaged NPs and CD47 functionalization are being developed for drug delivery. Unlike synthetic polymer-coated stealth NPs, which are dependent on hydration coronas, these alternative NPs possess active biological components on their surfaces for in vivo anti-phagocytic effects.

### 4.1. Cell Membrane-Coated Nanoparticles

In recent years, cell membrane cloaking has emerged as a prominent bio-stealth technique due to the intrinsic biological functions of the membrane-anchored proteins, as well as immunological moieties. Owing to the smart functions, particularly intercellular communication, bioantifouling as well as immune defense, the cell membranes have been isolated and extruded for NP fusion to mimic a cell-like behavior [83,84,85,86]. Red blood cells, platelets, immune cells, cancer cells, and even *Escherichia coli* have been exploited as membrane sources to develop biohybrid stealth systems with versatile functions [87]. Various NPs such as PLGA, liposomes, and gold NPs have been cloaked with natural cell membranes to enhance their targeting ability and circulation time for cancer therapies [88].

Red blood cells (RBCs), as the oxygen delivery carriers, were widely employed as bio-stealth materials to escape immune recognition and improve blood circulation half-life. In a study by Gao et al. [89], gold NPs were enclosed with cellular membranes of natural RBCs through a top-down approach. The RBC membrane-coated gold NPs not only effectively shielded the particles from thiolated probes, but also bestowed immunosuppressive properties for evading macrophage uptake. Hu et al. [90] developed a core-shell nanocarrier by coating PLGA NPs with the bilayered RBC membranes associated with both lipids and surface proteins. The resulting RBC membrane-coated polymeric NPs exhibited a significantly longer elimination half-life of 39.6 h as compared to that of PEG-coated NPs, which was calculated as 15.8 h in a mouse model. These findings indicate that RBC membrane-camouflaged NPs exhibit a prolonged blood circulation time by evading immune surveillance with their biomimetic features.

Macrophages, as the “lifeguards” of immune system, have proved useful for membrane coating in improving the circulation time of mesoporous silica nanocapsules in vivo, and enhancing the drug-delivery efficiency with improved tumoritropic accumulation as compared to uncoated NPs [91]. Interestingly, cancer cell membranes that possess unique features including cell death resistance, immune escape as well as long circulation time, have also attracted substantial interest as coating biomaterials for NPs. Recently, Sun et al. [92] fabricated a biomimetic drug-delivery system composed of doxorubicin-loaded gold nanocages as the inner cores, and 4T1 cancer cell membranes as the outer shells. This nanodrug-delivery system exhibited superior targeting efficiency and higher accumulation in tumor sites. The hyperthermia-triggered drug release also effectively suppressed tumor growth and metastasis of breast cancer.

Overall, the biomimetic functionalization through cell membrane cloaking is an innovative strategy to produce bioinert NPs for a variety of applications including drug delivery, phototherapy, and imaging applications. Membrane coatings mimic source cells, and mark NPs as “self”. The biological properties of cell-membrane coated NPs including immune evasion, prolonged circulation, and increased targeting capability, have significantly enhanced their potential in drug delivery applications. Further modifications with the incorporation of ligands including antibodies, peptides, enzymes, and proteins, will endow a new strategy in biomimetic platforms with enhanced synergistic performance [87,88].

### 4.2. CD47 Functionalization

In addition to cell-membrane camouflaged NPs as stealth delivery vehicles, CD47, a transmembrane protein that functions as a universal molecular “marker-of-self”, has attracted increasing interests for the development of bioinert immune-evasive biomaterials and NPs. This is attributed to the anti-phagocytic properties of CD47 through an inhibitory action via signal regulatory protein alpha (SIRPα) expressed on the macrophage membrane [93]. As illustrated in Figure 3, the stimulation of SIRPα by CD47 ligand negatively regulates phagocytosis, and produces a “do-not-eat-me” signal transduction on macrophage membrane [94]. The capability of CD47 in inhibiting phagocytosis and conferring anti-inflammatory properties have significantly contributed to the in vivo survival of RBCs [95], cancer cells [96], and viruses [97]. However, owing to the large size and complexity of protein folding, short-chain CD47-mimicking peptides are usually preferred in comparison to the recombinant proteins as the peptides confer higher biocompatibility, and facilitate chemical bonding of peptides to the surface of NPs [98]. The detailed mechanism of CD47/SIRPα regulation on phagocytosis has been described by Yang et al. [99].

In recent years, applications of CD47-mimicking peptides as stealth functionalization have been demonstrated to produce macrophage-evading NPs with increased blood circulation half-life and delivery efficiency. In a study by Rodriguez et al. [100], a CD47-mimicking peptide known as a ‘self’ peptide (SP) that specifically binds to, and signals phagocytes to impede the clearance of particles, was synthesized and attached to nanobeads for paclitaxel delivery. These CD47-functionalized nanobeads enhanced the antitumor efficacy with prolonged in vivo circulation and drug retention in tumors while exhibiting anti-phagocytic effects. Shim et al. [101] modified graphene oxide (GO) nanosheets with a CD47 mimicry SP and PEG to determine the phagocytic uptake by macrophages. The results showed that SP-coated GO nanosheets exhibited a greater distribution and accumulation to tumor tissues than the PEGylated GO nanosheets as the immune-camouflaging GO nanosheets reduced the clearance by macrophage uptake, rendering a prolonged blood circulation and effective delivery in vivo even after repeated administrations.

The anti-phagocytic properties of CD47-mimicking SP were also demonstrated recently by Jiang et al. [102] in reducing the macrophage uptake of biodegradable photoluminescent waterborne polyurethane polymer micelles. The novel SP-functionalized micelles diminished the particle accumulation in the liver and kidney while improving the targeting and retention in tumor sites. To date, this innovative approach of stealth CD47 functionalization has demonstrated prominent results at the experimental stage. The nanocarriers camouflaged as “self” can readily evade the immune surveillance despite their interactions with biological components in the circulation. Although this field is still in nascent development, these pioneering studies are paving the way to the clinical use of CD47-functionalized biomaterials and NPs, for targeted drug delivery in the near future.

## 5. Coating Characteristics for Stealth and Long-Circulating Nanoparticles

There are several key factors that play important roles in developing long-circulating polymer-coated stealth NPs in the bloodstream. Yet, it should be noted that certain parameters can cause inadvertent changes; for instance, increasing PEG molecular weight often leads to changes in the size, surface density, and stability of NPs in the blood [44,103]. Thus, it is extremely challenging to control precisely the effects of polymers’ coating characteristics on the surface of NPs due to the correlation of coating parameters. Since the development of PEG has led to a proliferation of studies as a stealth polymeric delivery system, here we discuss in detail some of the critical coating factors, including molecular weight, surface chain density, and surface conformation that influence the interactions and circulation of the PEGylated NPs in the blood.

### 5.1. Molecular Weight

Since the molecular weight (MW) of grafted PEG chains is proportional to its chain length, the PEG MW plays a key role in developing effective surface shielding [19]. It has generally been shown that PEG MW of at least 2 kDa is required to achieve efficient shielding from protein adsorption as shorter PEG chains are less flexible [30].

Fang et al. [104] fabricated different sizes (80, 170, and 240 nm) of NPs loaded with recombinant human tumor necrosis factor-α (rHuTNF-α) with three different MW (2, 5, and 10 kDa) of methoxypolyethyleneglycol (MePEG). The results demonstrated that the adsorption of serum proteins and phagocytic uptake were significantly decreased with higher MW of MePEG or a smaller size of NPs, leading to prolonged drug accumulation in tumors and extended circulation time in vivo. This is due to the fact that higher MePEG MW resulted in a thicker fixed aqueous layer thickness (FALT), whereas a smaller particle size offered a higher MePEG chain density. Likewise, another study by Cui et al. [105] also found that by increasing PEG MW (from 10 to 40 kDa) or decreasing the PEG particle size (from 1400 to 150 nm) led to reduced phagocytic blood cell association of the particles. The in vivo biodistribution studies revealed that the smaller PEG particles exhibited extended circulation times (>12 h) in comparison to larger PEG particles. Despite the controlled PEG particle size in these studies, it was unclear whether the surface density of the grafted PEG chain remained constant while increasing the PEG MW. Gref et al. [44] evaluated plasma protein adsorption of PEG-coated PLA NPs using different PEG MW ranging from 2 to 20 kDa. In contrast to other studies, the results showed that protein adsorption was decreased significantly when PEG MW was increased from 2 to 5 kDa, but no further reduction of plasma protein adsorption was observed when PEG MW was increased above 5 kDa. It was thought that the protein adsorption on the surface of NPs could not be prevented as they were not fully protected by the grafted PEG chains, rendering the interactions between plasma proteins and NPs feasible. Contrary to the claim that shorter PEG with low flexibility cannot reduce NPs’ uptake [30,103], Yang et al. [99] reported that the surface of 100 nm polystyrene NPs could be effectively shielded by 559 Da PEG, when the grafting density exceeded 1.2 PEG/nm^2^ [106]. Therefore, it is speculated that grafting lower MW PEG with higher surface density may exhibit similar effectiveness in extending circulation time as grafting higher MW PEG on the surface of NPs.

### 5.2. Surface Chain Density

Besides PEG MW, it is also recognized that the surface chain density of the grafted PEG is an associated key factor that influences the stealthiness of NPs [11]. The grafted polymer chains must cover the entire surface of NPs to provide efficient shielding from the surrounding plasma proteins [107,108]. Indeed, several studies emphasized the importance of polymer density as high surface density could compensate low polymer MW by means of preventing the protein adsorption on the surface of NPs [46,108,109]. In other words, when the surface density is too low, the NPs surface associated with charge and hydrophobic moieties will become more accessible to surrounding plasma proteins [46].

Recently, Shalgunov et al. [52] conducted an in vivo study to evaluate the influence of PEG coverage on the biodistribution of vincristine encapsulated into PLA-PEG NPs. They found that NPs with lower PEG coverage exhibited rapid elimination from the bloodstream following intravenous injections. However, an increase in PEG coverage was shown to improve NPs circulation half-life efficiently and decreased their retention in the liver, spleen, and lungs. Additionally, they demonstrated that NPs with lower PEG coverage increased the uptake by macrophages in vitro, further justifying the importance of surface chain density in shielding NPs against the macrophages. Contrary to the study, another in vivo biodistribution study by Peracchia et al. [43] investigated the impact of PEGylation coating degree by means of extended blood circulation half-life of PEG-coated polyalkylcyanoacrylate (PACA) NPs. Even though a substantial increase in circulation was pronounced for PEGylated PACA NPs, an increase in the PEG content in the NPs’ synthesis did not show further augmentation of the circulation time. This might due to the fact that an optimal PEG surface density was already achieved for efficient steric repulsion to occur. These results further suggest that by increasing PEG content may not provide higher surface coverage of PEG to reduce the protein adsorption on the surface of NPs.

The MW and surface chain density are two interrelated criteria which can compensate each other in creating an optimal coating thickness layer for protein repulsion. Yet, the distinct relationship between PEG MW and its surface density can cause a variation in coating effects under different experimental quantifications and analyses. However, there exists a density limit where protein adsorption cannot be decreased with no apparent extended NP circulation time in the blood. In addition, it should be noted that the polymer density threshold critically depends on several crucial factors, in particular, particle size and its surface curvature.

### 5.3. Surface Conformation

In general, among the parameters that influence NPs’ stealthiness, surface chain density and conformation are more correlated for effective particle protection. At low surface coverage, the grafted PEG chains exhibit larger range of motion, and result in a “mushroom” configuration where most of the chains are located closer to the particles’ surface. Moreover, low surface coverage of NPs will create spaces that cause opsonins to interact with NPs for elimination. On the other hand, the PEG chains’ range of motion will be substantially constrained at high surface coverage, creating a “brush” configuration on the surface [30]. Even though high surface coverage bestows effective shielding for the entire surface of NPs, it is thought that a high coverage may reduce the mobility of PEG chains and their steric hindrance properties [110]. Therefore, it is a necessity to achieve an optimal surface coverage between both “mushroom” and “brush” configurations, where the grafted chains possess a certain range of motion at surface density that is high enough for protein repulsion.

PEGylation density is often described in terms of surface conformation of the adopted PEG chains as presented in Figure 4, which is based on the Flory radius (*R_F_*) of the PEG graft, the distance (*D*) between PEG graft, or the thickness (*L*) of the grafted PEG layer [111]. At low grafting density, where *D* > *R_F_* and *L* = *R_F_*, PEG chains adopt a “mushroom” conformation, and they are not fully extended away from the particles’ surface. On the other hand, “brush” conformation is dictated by having an increased grafting density, where *D* < *R_F_* and *L* > *R_F_*, with PEG chains fully extending away from the particles’ surface [112,113]. The *R_F_* and *D* can be determined using the following equations: *R_F_* = *aN*^3/5^, *A* = 1/*P*, and *D* = 2(*A*/*π*)^1/2^; where *a* is the monomer length of the PEG chain (0.35 nm), *N* is the degree of polymerization (number of PEG repeats), and *A* is the area occupied per PEG chain. The mushroom and brush conformations are dictated by *R_F_*/*D* ≤ 1 and *R_F_*/*D* > 1, respectively [111]. Since the transformation from mushroom to brush conformation depends on the surface concentration, Damodaran et al. [112] proposed an assignment of dense brush conformation only when the layer thickness is significantly greater than the *R_F_* value, when *L* > *2R_F_*. Thus, these key parameters: *D*, *R_F_*, and *L*, are utmost important in distinguishing the possible conformation of the PEG chains.

Yang et al. [106] conjugated 5 kDa PEG chains to the surface of 100 nm polystyrene NPs. They found that a dense brush conformation with surface PEG densities considerably exceeding the mushroom-brush transition threshold (*R_F_*/*D* = 1) is a requisite for effective suppression of macrophage uptake. These PEGylated formulations having PEG coatings with *R_F_*/*D* ≥ 2.8 showed a dense brush regime on the surface of NPs. For longer PEG chains (≥10 kDa), a substantially higher PEG density (*R_F_*/*D* > 8) is required for maximal macrophage uptake suppression. In addition, the in vivo blood circulation and biodistribution profiles revealed that particles with brush (*R_F_*/*D* = 2.0) and dense brush (*R_F_*/*D* = 4.2) regimes were cleared within 2 h, leading to rapid accumulation in the liver as a result of MPS clearance. However, highly dense PEGylated particles (*R_F_*/*D* ≥ 6.6) showed a prolonged blood circulation half-life of 14 h, with eventual accumulation primarily in the liver. In contrast to the aforementioned study, Perry et al. [113] coated hydrogel NPs with 5 kDa PEG at varying grafting concentrations. As a result, they observed that both PEGylated hydrogel NPs in mushroom (*R_F_*/*D* ~ 0.9) and brush (*R_F_*/*D* ~ 1.5) conformations exhibited comparable reduced uptake by macrophages. Interestingly, the blood circulation half-life was substantially increased from 0.89 h for uncoated hydrogel NPs to 15.5 and 19.5 h for PEGylated hydrogel NPs with mushroom and brush conformations, respectively. This inconsistency could be attributed to the presence of PEG in the NPs’ core, and the soft mechanical structure of the hydrogel NPs.

To evade the protein adsorption through gaps or spaces effectively, the entire particle surface needs to be grafted with sufficient PEG chains at a very dense brush surface density. In light of the long-held notion that surface coating with brush conformation confers efficient stealth properties to NPs, it should be worth noting that the *R_F_*/*D* threshold is dependent on PEG MW and the NPs’ size. Additionally, different methods of PEG-coating quantification remain as a critical hurdle to improve PEGylation for drug-delivery systems. Therefore, it is essential to develop a superior and reliable quantification assay to gain a detailed understanding of the surface-grafted PEG chains.

## 6. Conclusions

Over the years, the development of polymer-coated NPs has led to significant advances in developing nanocarriers with the ability to evade the immune surveillance for stealth drug delivery. Although PEGylation remains the most widely used strategy for MPS-avoidance characteristic, other stealth-functionalization approaches such as cell-membrane camouflaging and CD47 mimicry have shown great potential in reducing the protein adsorption and increasing blood circulation times of NPs. However, it should be noted that the invention of long-circulating stealth NPs is not a facile but complex task that requires detailed understanding of the interactions between NPs and the biological environments. Several critical coating parameters, including MW, surface density as well as the surface conformation, can influence the stealthiness of polymer-coated NPs. While extensive research has emerged in the field of stealth NPs for targeted drug delivery, further studies are required to evaluate the effectiveness and safety of the stealth NPs for clinical applications. By successively addressing the formidable challenges with in-depth understanding, long-circulating stealth nanotherapeutics for site-specific delivery will soon become a reality.

## Figures and Tables

**Figure 1 nanomaterials-10-00787-f001:**
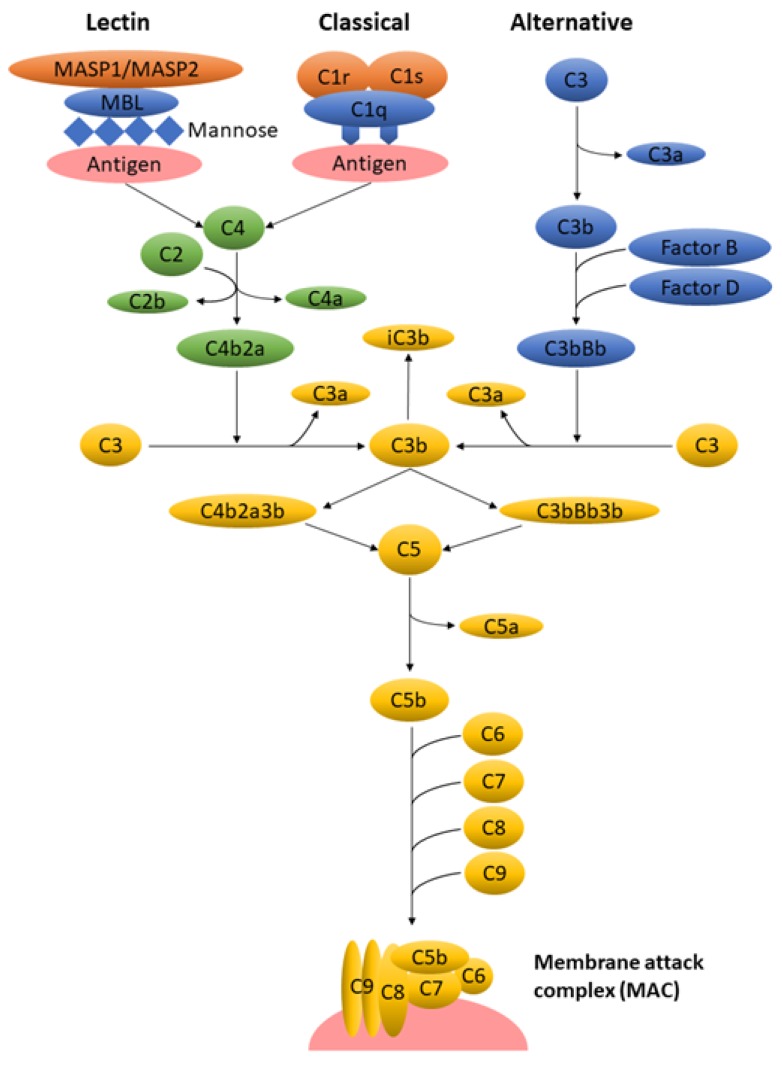
A simplified overview of different activation pathways of the complement system. There are three complement activation pathways: the classical pathway, which is activated by antibody binding or the direct fixation of complement component C1q bound to zymogens C1r and C1s on the surface of an antigen; the lectin pathway, which is triggered by the binding of mannan-binding lectin (MBL) activated by MBL-associated serine proteases (MASP), namely MASP1 and MASP2, to mannose contained on the surface of an antigen; and the alternative pathway, which is triggered directly by the binding of spontaneously activated complement component on the surface of an antigen. The complement enzymatic cascades of each pathway generate a key protease called C3 convertase that cleaves C3 into C3b and C3a. This complement activation leads to eventual antigen opsonization, inflammatory responses, and membrane lysis.

**Figure 2 nanomaterials-10-00787-f002:**
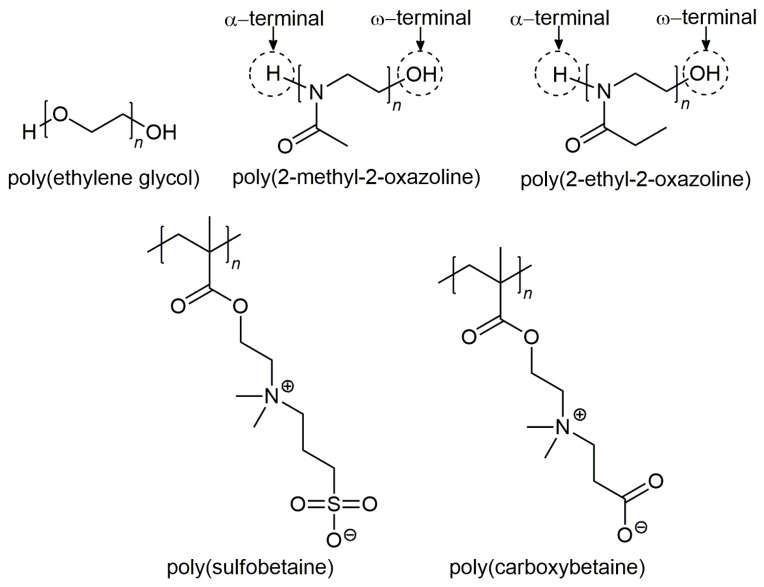
Chemical structures of the stealth polymers. The alpha (α) and omega (ω) termini of poly(2-methyl-2-oxazoline) and poly(2-ethyl-2-oxazoline) are indicated in circles.

**Figure 3 nanomaterials-10-00787-f003:**
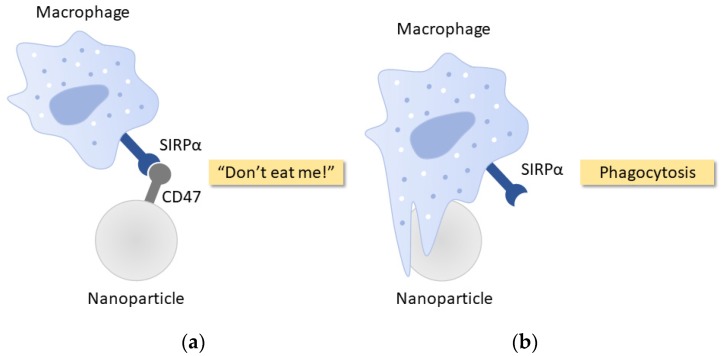
Schematic representations of CD47 regulation on phagocytosis of nanoparticles (NPs). (**a**) CD47 coated on a nanoparticle interacts with the signal regulatory protein alpha (SIRPα) expressed on the surface of the macrophage, triggering a potent “don’t-eat-me” signal, which inhibits phagocytosis; (**b**) A nanoparticle without CD47 functionalization is recognized by macrophage for particle engulfment and phagocytosis.

**Figure 4 nanomaterials-10-00787-f004:**
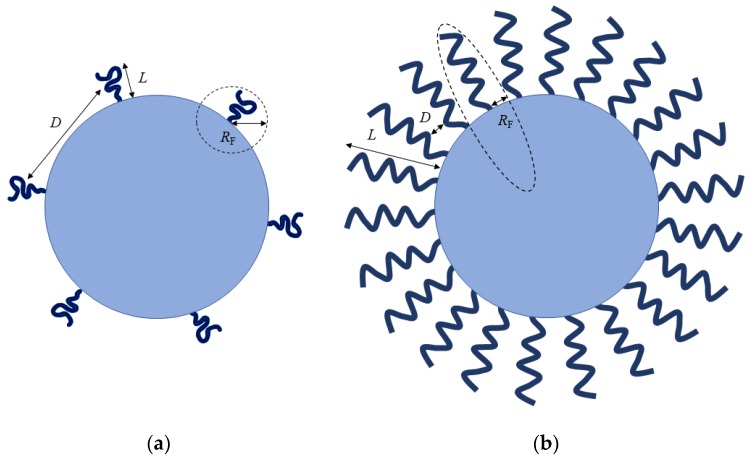
Schematic representations of poly(ethylene glycol) (PEG) conformations on NPs. (**a**) At low surface coverage, PEG chains are located closer to the particle’s surface, leading to a mushroom conformation; (**b**) At high surface coverage, PEG chains are lack of mobility and extended away from the particle’s surface, leading to a brush conformation. *R_F_* represents the Flory radius of the PEG graft; *D* represents the distance between the adjacent PEG grafts; *L* represents the thickness of the grafted PEG layer (the diagrams are drawn not to scale).

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
