# Peer review of "Stealth Coating of Nanoparticles in Drug-Delivery Systems"

_nanomaterials, 2020, doi:10.3390/nano10040787_

Round 1

Reviewer 1 Report

This is a delightful review. By and large it is well written and right to the point. The only criticism I have of the manuscript is that it is somewhat sparse on figures. It may be too late at this point; however, perhaps as Fig. 2, an additional figure could be added that illustrates the chemical functional groups involved in the polymers listed in sections 3.1, 3.2 and 3.3. Perhaps also, if available, a more detailed depiction of CD47 could be added either as a stand-alone figure or as another panel in the current Fig. 2. After consideration of these points and correction of the minor items listed below, I believe this manuscript will be suitable for publication.

There are just a few editorial corrections that I would suggest:

Lines 41 and 162: The more concise “Undoubtedly” would be better than “It is undoubtedly that”.

Line 142: Change to “there are no definite rules”.

Line 180: Change to “it is a necessity”.

Line 274: Change to “solvation”.

Line 275: Change to “creating highly hydrophobic surfaces”.

Line 418: Change to “contrary to the claim”.

Line 465: Change to “it is a necessity”.

Author Response

Please refer to the attachment for point-by-point response to Reviewer 1 comments.

Reviewer 2 Report

The authors describe in this review article the different approaches for stealth coating of nanoparticles for improving blood circulation half-life of the nanomaterials. Surface coating characteristics of PEGylated NPs are also discussed. In my opinion the manuscript can be accepted for publication after some revision:

  1. Two recent reviews on stealth coating of nanoparticles should be included:

- Z. Amoozgar, Y. Yeo. Recent advances in stealth coating of nanoparticle drug delivery systems. WIREs Nanomed Nanobiotechnol 2012, 4, 219-233.

- N. Hadjesfandiari, A. Parambath. Stealth coatings for nanoparticles: Polyethylene glycol alternatives. From the book: Engineering of biomaterials for drug delivery systems. Beyond poliethylene glycol. Woodhead Publishing Series in Biomaterials, 2018, pages 345-361.

  1. There is a lack of Figures (there are only 3 Figures in the manuscript), which makes the review difficult to read. For instance, it would be convenient to add figures with the chemical structures of the different polymers (PEG, Poly(2-methyl-2-oxazoline), the alpha and omega termini of POx, the structures of poy(zwitterions) PSB and PCB or an example of nanoparticles covered by some of these stealth polymers.
  2. Regarding the cell membrane-coated NPs, an explanation of how this cell membrane cloaking is performed is missing in the text.
  3. Finally, in Figure 3, I don’t see any difference between the mushroom model in the left and the brush model in the right. In both models the grafted PEGs look the same…

Author Response

Please refer to the attachment for point-by-point response to Reviewer 2 comments.
